# On Robustness of Multi-Modal Fusion—Robotics Perspective

**Michal Bednarek \*** 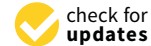**, Piotr Kicki and Krzysztof Walas**

Institute of Robotics and Machine Intelligence, Poznan University of Technology, 60-965 Poznan, Poland;
piotr.kicki@put.poznan.pl (P.K.); krzysztof.walas@put.poznan.pl (K.W.)
**\*** Correspondence: michal.bednarek@put.poznan.pl

**Abstract:** The efficient multi-modal fusion of data streams from different sensors is a crucial ability that a robotic perception system should exhibit to ensure robustness against disturbances. However, as the volume and dimensionality of sensory-feedback increase it might be difficult to manually design a multimodal-data fusion system that can handle heterogeneous data. Nowadays, multi-modal machine learning is an emerging field with research focused mainly on analyzing vision and audio information. Although, from the robotics perspective, haptic sensations experienced from interaction with an environment are essential to successfully execute useful tasks. In our work, we compared four learning-based multi-modal fusion methods on three publicly available datasets containing haptic signals, images, and robots' poses. During tests, we considered three tasks involving such data, namely grasp outcome classification, texture recognition, and—most challenging—multi-label classification of haptic adjectives based on haptic and visual data. Conducted experiments were focused not only on the verification of the performance of each method but mainly on their robustness against data degradation. We focused on this aspect of multi-modal fusion, as it was rarely considered in the research papers, and such degradation of sensory feedback might occur during robot interaction with its environment. Additionally, we verified the usefulness of data augmentation to increase the robustness of the aforementioned data fusion methods.

**Keywords:** multi-modal fusion; machine learning; robotics

## 1. Introduction

A dynamic fusion of multi-modal information is a key ability that humans utilize for a wide variety of tasks that demand an understanding of the physical properties of objects. For example, we fuse visual and haptic data to manipulate dexterously, recognize unknown objects, or localize them in the scene. However, when we want go to the bathroom at night and we want to open the water tap, somehow we know that an image under these conditions is not reliable and we should focus more on our other senses. By touching the tap we can (to some extent) supersede the vision in the localization task and perform the same task without it. The interaction between information from different senses was observed and tested experimentally in [1], where a multi-sensory illusion was called the McGurk effect. In the robotics field, a typical approach to the multi-modal data fusion is through various probabilistic models that are based mainly on the Bayesian inference. However, due to large volumes of available multi-modal and multi-relational datasets, that kind of analysis can be hindered. To overcome that problem machine learning approaches were proposed, as they can handle large and multidimensional data. In recent years there was a lot of research in the area of efficient fusion of data using machine learning, especially neural networks [2]. Nevertheless, researchers focused on the improvements in the accuracy of their models and paid almost no attention to their robustness to non-nominal conditions, which are ubiquitous in the robotics applications.

To bridge this gap, in our work we compared the performance of four popular multi-modal fusion methods utilizing artificial neural networks (ANN) on three datasets. Moreover, we extensively tested them in various data degradation scenarios, which simulated the typical issues like noisy data or sensor failure. Furthermore, we verified the influence of data augmentation on a fusion methods robustness. The general design of our experiments was depicted in Figure 1.

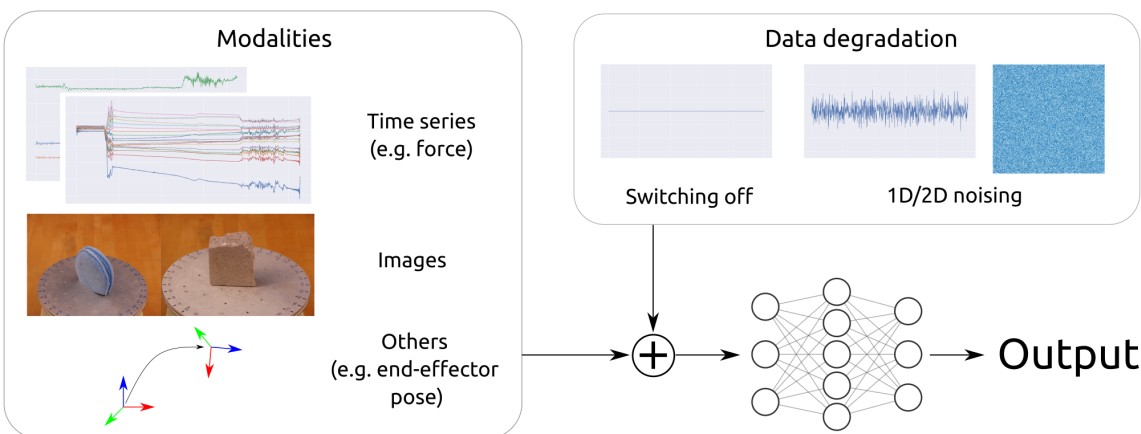

**Figure 1.** Experimental setup. We tested four different methods of multi-modal fusion and verified their robustness on a variety of data degradation scenarios common in robotics.

The development of approaches for fairly complicated robotic tasks, like, e.g., dexterous manipulation, nowadays very often depends more on advances in sensory systems, such as skin-like sensors [3–5] than fusion algorithms themselves. The use of fusion in dexterous manipulation [6,7] is emerging field. The more focus on development of multi-modal systems in the field of robotics is especially visible in areas like image segmentation [8–10], 3D reconstruction [11] and a tactile understanding [12].

In our work we want to stress the fact that in robotics, a multi-modal fusion is particularly important due to a possibility to improve the robustness of predictions affected by noise or failure of a sensor. Typically, it can be achieved by finding pieces of information among different modalities that exhibit interchangeability and complementarity. However, in most applications researchers focuses mostly on the improving the accuracies of their systems, by exploiting the complementarity. There is almost no consideration on interchangeability of the data sources, in the context of the robustness to, typical for real-life scenarios, noises and sensor faults.

A multi-modal machine learning is a scientific field of growing interest that brings many challenges. In [2] the authors have listed open questions to which answers should be found to advance the state of the art—how to represent the data (representation), how to map a knowledge from one modality to others (translation), how to find dependencies between heterogeneous modalities (alignment), how to join the multi-modal data stream together (fusion) and how to successfully transfer a knowledge from training a model on one modality to other (co-learning). We are aware that there are many more data fusion techniques like Kalman Filter, Bayesian Inference or Early Fusion. Unfortunately, each of those solutions is somewhat limited to low-dimensional or homogeneous data, thus we decided to exclude them from our comparison and focused on the most flexible approaches, which are model agnostic and can operate on any type of data. In our work, we evaluated a performance of multiple data fusion methods based on neural networks in robotics oriented tasks. Our contributions are:

1. Experimental evaluation of the performance of four machine-learning fusion methods—Late Fusion (Late), Mixture of Experts (MoE) [8], Intermediate Fusion (Mid) and the most recent Low-rank Multi-modal Fusion (LMF) [13]. We tested their capabilities on three multi-modal datasets in single and multi-label classification tasks.

2. Validation of the robustness of selected fusion methods to various data degradation scenarios that may occur when the robot interacts with the environment.
3. Evaluation of the influence of a data augmentation on a fusion methods performance, when a data degradation occurred during tests.

The remainder of the paper is organized as follows. First, we will provide a comprehensive review of the related work in the field of multi-modal data fusion. Then, we will present tested fusion methods and datasets used in our experiments. Next, we will move on to the results section followed by the discussion. Finally, concluding remarks will be given.

## 2. Related Work

The section contains a broad review of research conducted on multi-modal fusion approaches and their evaluation for data degradation, which might occur in real-world scenarios.

### 2.1. Data Fusion Approaches

In [2] the authors divided the multi-modal fusion techniques into two classes: model agnostic and model-based. In our paper we limited ourselves to model agnostic fusion methods as they are more general and widely spread in the robotics community. There exist three main types of model agnostic data fusion methods—early (data-level), intermediate (feature-level), and late (decision-level) [14], however it is possible to combine at least two of them into a hybrid fusion [2]. Systematic division of different sensor fusion methods is given in [15].

In early fusion, data from different modalities are typically concatenated at the early stages of data processing. It gives the machine learning model a possibility to capture even low-level interactions between modalities and process them jointly. However, that approach is limited to the cases when it is possible to concatenate the data, which is sometimes cumbersome. For example, it is not clear how to combine heterogeneous data such as 2D images with a 1D time series. For that reason, we do not compare early fusion in this paper, as it is not applicable to all data types. Typical examples of early fusion can be found in the area of semantic scene understanding using multi-spectral images [16], where visual streams from RGB, depth, and near-infrared channels are combined to produce predictions. Another example of an early fusion approach was [17]. The authors proposed to fuse RGB images with optical flow maps for gesture recognition. Other example is the use of different depth sensing modalities, with different properties to obtain denser depth image [18].

A feature-level fusion is a very popular technique in machine learning models as it merges data representations at higher levels of abstraction. That in turn allows for combining even heterogeneous data from very different sources and lets the machine learning model to process joint representation in order to produce an output. That type of fusion is widely spread in robotics community in areas such as object recognition [19] and scene recognition [20] tasks. Multi-modal fusion applied to robot motion planning was presented in [21] and contact-rich manipulation tasks in [7]. A different approach to a feature fusion is presented in [22], where instead of features concatenation, only some randomly chosen parts of feature vectors from different modalities are merged. Another intermediate fusion approach exploits Tensor Fusion Networks [23,24], which are extensively used for example in the multi-modal sentiment analysis. However, they, to the best of the authors' knowledge, are not used in the robotics applications. The main issue of these approaches is their low computational efficiency, which is addressed in the paper about Low-rank multi-modal Fusion [13], which exploits the tensor decomposition to reduce the number of model's parameters.

Similar to feature-level fusion approaches, late ones can work with any data types. However, they do not merge the data but only outputs of the models, which process different modalities separately. Prominent work on late fusion approaches is described in [25]. The typical scheme of a late fusion was presented in [26–28]. The authors of [27,28] proposed the late fusion approach to process RGB-D data in the tasks of object detection and discovery respectively. In [26] images and

point clouds were used to perform semantic segmentation of the urban environment for autonomous vehicles. In [29] the authors proposed a late fusion model, which took into account the impact of a data degradation on the model's decision and used a noisy-or operation to combine these decisions. The use of three different modalities for terrain classification fused using late fusion approach is presented in [30].

If the proposed solution merges information on two or more levels, we talk about hybrid fusion. Examples of that approach are presented in [8,9] where outputs of models are fused with the use of weights determined by the gating network, which uses an intermediate representation of all modalities. That approach, called the Mixture of Experts is able to decide, which modality should have a stronger impact on the final outcome based on the features extracted from all modalities.

### 2.2. Fusion Robustness

Fusion robustness is a rarely considered topic in the multi-modal fusion literature, however, it seems to be an important issue in real-world applications, especially in the robotics field. Only several papers [22,29,31–34] took into account the non-nominal conditions of the multi-modal fusion and provided some analysis of fusion robustness to data degradation. Such degradation could occur due to sensor noise, its failure, or unexpected weather conditions. Moreover, the approach proposed in [8] is potentially able to take into account data degradation and express the belief in terms of the weighting of the model's decisions. However, it was not considered by the authors. Even though there are works on the robustness of multi-modal fusion in the robotic context, one can observe a lack of comprehensive joint comparison of main fusion paradigms, which were used by individual authors in the presence of sensor noises and/or failures.

## 3. Experiment Design

In the following section, we provided a detailed description of multi-modal datasets used in our experiments including the procedures of data preparation, cross-validation and splitting into train/test subsets. Moreover, we presented compared fusion methods with a discussion of their architectures.

### 3.1. Data Preparation

In our work, we measured the performance and robustness against data degradation of different fusion methods using three datasets containing multi-sensory data. First of all, from each dataset a test subset was separated and it remained unchanged throughout all of our experiments. The important note is that the test set was not involved in any training procedures described further and served only as a reference point for comparisons between methods. To ensure a fair comparison, for each dataset, we ensured that the distribution of the classes both for train and test set is similar. The same principle was maintained for the cross-validation. Each class was evenly distributed between consecutive folds using iterative stratification method [35,36]. Thus the case that some class was under or over-represented in some part of data was eliminated.

In our work, one turn of cross-validation proceeded as follows. We split the dataset into $k$ chunks called folds. Folds numbered from 0 to $k-1$ were used for training an ANN. After that, the $k$-th fold was used for validation. That procedure was repeated 5 times, each time different fold was chosen as the validation set (5-fold cross validation). Moreover, in all our experiments, input data was standardized by subtracting the mean taken for all samples from a corresponding modality and divided by its standard deviation.

### 3.2. Multi-Modal Datasets

BioTac Grasp Stability Dataset (BiGS): The grasp-stability dataset [37] contained signals recorded during 2000 trials of grasping three types of objects—a ball, box, and a cylinder. Time series of gripper's poses and 3-axis forces gathered while shaking an object with a closed gripper. Each trial was annotated with a label *success* or *fail* that corresponds to the outcome of a trial. To gather tactile signals

there were used three bio-inspired BioTac [5] tactile sensors mounted on fingers of a gripper and a force-torque sensor mounted in the wrist. In our experiments, to verify whether a signal represents a successful grasp or a failure, we used gripper's positions, orientations, and force readings from the force-torque sensor. We re-sampled each signal to fit common length using the Fourier method. Finally, each input element consisted of three-time series with a length equal to 1053. Positions are expressed as 3-element vectors, orientation is presented as a 4-element quaternion, and force reading is composed of 3 values corresponding to X, Y, and Z axes. In total, the training dataset used in the cross-validation was composed of 3197 samples for each modality, while the test set of 801.

The Penn Haptic Texture Toolkit (HaTT): In the toolkit [38] there are 100 different textures photographed and presented as RGB images. Each texture has associated normal force, acceleration, and position recordings gathered during unconstrained motions of an impedance-type haptic device SensAble Phantom Omni [39]. To save time needed for training, in our experiments we used data from 10 classes only, the names of which were presented in Table 1.

**Table 1.** Textures from the Penn Haptix Toolkit chosen for experiments.

| ABS Plastic | Aluminium Foil | Aluminium Square | Artificial Grass | Athletic Shirt |
|---|---|---|---|---|
| Binder | Blanket | Book | Brick 1 | Brick 2 |

Signals in the HaTT dataset were gathered using a haptic device's tool-tip while moving on different surfaces for 10 s. In our experiments, we used a normal force, acceleration and velocity as input modalities. However, it is important to mention that the authors of the dataset used a method called DFT321 [40] to combine 3-axes signals of acceleration and velocity into single axes, thus in our experiments, we used the 1-dimensional representations of these quantities. The authors motivated that dimensionality reduction by the fact that humans do not perceive the direction of high-frequency vibrations, which was described in [41]. We did not use available RGB images, because each class had only one associated image, thus there were far too few of them to train an ANN. Every time series was cut into vectors with a length of 200, which resulted in a total number of 8000 samples included in the training set and another 2000 in the test set. Again, we made this split maintaining the equal balance between classes.

The Penn Haptic Adjective Corpus 2 (PHAC-2): The last dataset used in our experiments considers the problem of multi-label classification of haptic adjectives using data created by the authors of [42] and dataset was further refined by authors of [12]. The dataset consisted of 53 objects photographed from 8 different directions. Each photo had corresponding haptic signals from the squeezing of an object gathered from two BioTac sensors. Moreover, every object was described with several haptic adjectives used as labels. In the dataset there were 24 haptic adjectives, in Figure 2 we presented their histogram.

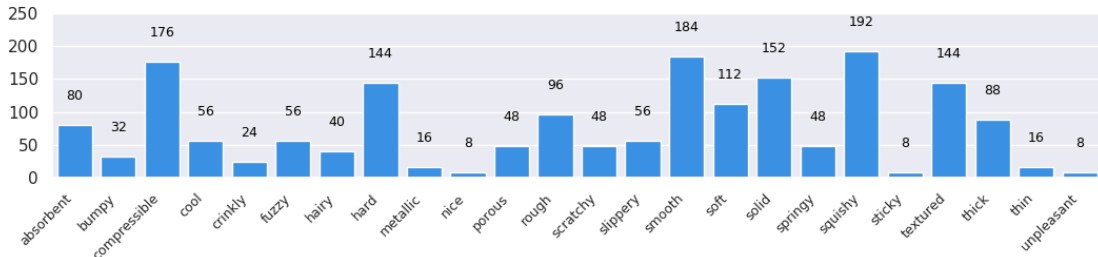

**Figure 2.** Occurrences of each adjective in the Penn Haptic Adjective Corpus 2 (PHAC-2) dataset.

To perform the experiments on the balanced train and test sets, we used the iterative stratification to ensure that there is no significant over or under-representation of any adjective in the train/test

subsets. It is important, because such imbalance can lead to a significant drop in the prediction performance, hence giving misleading results. A single training sample was composed of an RBG image with the spatial resolution of 224 × 224 together with two raw signals from 19-electrode arrays from both of BioTac sensors. As the length of time series was different, we re-sampled them to fit a fixed number of 67 values. In total, we had 265 samples in the training dataset and 159 in the test set.

### 3.3. Fusion Methods

In related work we discussed several model agnostic data fusion techniques, but for the experimental analysis we selected four of them (see Figure 3). Taking into account the simplicity and popularity we decided to include in our benchmark basic versions of late and intermediate fusion. The third fusion candidate was the Mixture of Experts [8] fusion model, which was similar to late fusion, but it was able to decide on the modality importance based on their latent representations. Moreover, we also included a method that was previously not used in the robotics community but achieved some very promising results in other areas such as sentiment analysis—the Low-Rank multi-modal Fusion (LMF). To obtain fair comparison, in our implementation of aforementioned fusion techniques we first transformed input modalities to the 10-dimensional latent space. In this way we obtained $N$ latent vectors $L_1, L_2, \ldots, L_N$, which were provided to corresponding ANNs. Approaches to data fusion, which we decided to examine in this paper, are presented schematically in Figure 3 and described in detail below.

Late Fusion: The main idea standing behind that method was to process each modality separately and merge predictions at the very end of the process assuming that they were of the same importance. The merging is performed on the decision level as it is described in [15]. Referring to Dasarathy classification this approach is described as Decision In-Decision Out (DEI-DEO). In our experiments we processed each of latent vectors separately using ANNs (represented in Figure 3 as arrows) to obtain predictions for each modality $p_1, p_2, \ldots, p_N$ in the form of logits. Next, those logits were summed up and transformed into class probabilities using a softmax function.

Mixture of Experts (MoE): The approach presented in [8] is built upon the Late Fusion method, however, it could decide on the modalities importance through the gating network. That decision was encoded in the weights vector $w$, such that $\sum_{i=1}^{N} w_i = 1$. In contrast to the Late Fusion, before a summation of predictions from all modalities they were multiplied by corresponding weights. Thus, the value of a vector $w$ was determined by a relatively small fully connected neural network, which used all latent vectors and produced final predictions. That architecture potentially allowed ANN to learn how to react to the degradation of some modalities, by assigning lower weights to the degraded modalities. On the other hand, if the data degradation did not occur during a training phase, there was a possibility that the MoE would put too much emphasis on the modality affected by some noise during testing, which might result in false predictions.

Intermediate Fusion (Mid): The fusion of information carried by individual modalities was made by concatenating their representations in the latent space. Next, a common representation was processed further to obtain a joint prediction. The merging is performed on the feature level as it is described in [15]. Referring to Dasarathy classification this approach is described as Feature In-Feature Out (FEI-FEO). In our experiments joint predictions were produced by ANN. That approach allowed a fusion model to take into account data from all modalities in the latent space and process them freely. The Mid method would also be able to gain some robustness to the data degradation during the training, as it could learn to reduce the impact of degraded modalities. However, in contrast to MoE, its robustness and decisions were not so clearly interpretable.

Low-rank Modality Fusion (LMF): In our work, we used also a method very different than others. It was the tensor-based approach for multi-modal fusion, which was focused on revealing the interactions between features extracted from different modalities. Generally, the core idea of tensor approaches is a creation of some high-dimensional tensor representation by taking the outer products over the set of uni-modal latent representations. That representation is then linearly mapped

to some low-dimensional space using learned weights and biases. Typically, such approaches suffer from computational inefficiency as that tensor weights and the number of multiplications scales exponentially with the number of modalities. However, the approach proposed in [13], did not multiply high-dimensional weight tensor with the tensor representation of the data directly. Instead, the authors proposed to firstly decompose tensor weights into $N$ sets of modality-specific factors similar to the representation of the input decomposes into low-dimensional feature vectors. Such decomposition reduced the number of computations, as it let to directly map from feature space to predictions without explicitly creating any high-dimensional tensors.

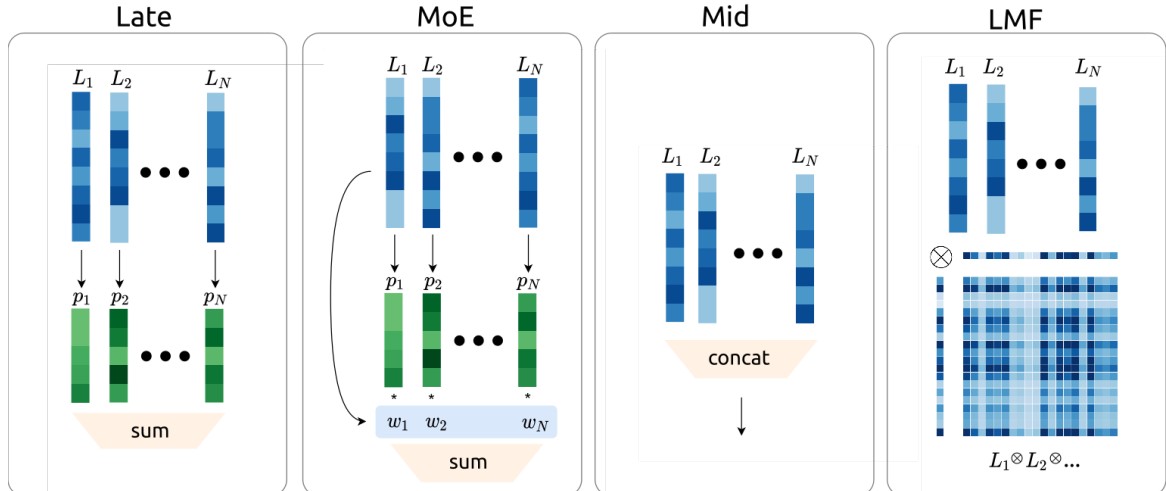

**Figure 3.** Multi-modal fusion architectures used in experiments. From left: Late Fusion (Late), Mixture of Experts (MoE), Intermediate Fusion (Mid), Low-rank Modality Fusion (LMF). Arrows represent the transformations realized with the use of neural networks, while $L_i, p_i, w_i$ denotes the latent vector, predictions and trainable weight associated with $i$-th modality.

### 3.4. Neural Network Architectures

To achieve a fair comparison between fusion methods, we designed neural network architectures in a way that ensured a similar number of the trainable parameters for each type of fusion. In our experiments, we operated on two different kinds of signals—time series and images. For time-series processing we used a few 1D convolutional layers (Conv1D), followed by the Long-Short Term Memory (LSTM) units and fully connected (FC) layers, whereas for images we used 2D convolutional layers (Conv2D) with a few FC layers on top of them.

For both BiGS and HaTT datasets, we used similar architectures for determining the latent vectors. They were composed of 3 Conv1D layers with 64 filters of size $5 \times 5$ with stride equal to 2, followed by LSTM layer with 32 units, and 2 FC layers with 128 and 10 neurons respectively. However, in the case of the BiGS dataset, for Mid, Late, and MoE the number of units in the last FC layer was changed to 2, as we performed binary classification. Moreover, for both BiGS and HaTT we reduced the number of filters in all convolutional layers for MoE, as it used an additional network to produce the weights $w_i$. This network for all datasets had the same architecture, namely, 3 FC layers with 128, 64, and $N$ units, where $N$ is the number of modalities. A similar network was used in the Mid fusion to process the concatenated latent vectors into the predictions, however, in the last layer, the number of units was equal to the number of classes.

In the case of PHAC-2 dataset, we had to process both time series as well as images. Neural networks for time series had similar architecture as for BiGS and HaTT datasets, however with an increased number of neurons in last FC layer—24 and reduced number of Conv1D layers equal 2 for all methods except the LMF, which stayed with 3 Conv1D layers and 10 neurons in the last FC layer. For image processing we used 2 Conv2D layers with 64 filters of size $5 \times 5$ with stride equal

to 2, followed by 2 FC layers with 128 and 24 neurons, except the LMF, which had 10 neurons in the last layer.

Additional details about the implementation of the fusion methods, as well as the architectures of the neural network used in the experiments, can be found in the code repository (https://bitbucket.org/m_bed/sense-switch/).

## 4. Results

In the following section, we presented results form the performance evaluation of four multi-modal fusion methods (Late, MoE, Mid, LMF) on three different datasets (BiGS, HaTT, PHAC-2). We firstly did 5-fold cross-validation (k-folds I-V) and presented results in Section 4.1. At that stage, we not only did the cross-validation but also chose the best performing models for further experiments. In tables, the chosen ones were marked with a blue color. After that, we measured the influence of multiple data degradation scenarios on the performance of each method and reported results in Section 4.2. Finally, the Section 4.3 contains the outcome of experiments conducted towards data augmentation and its influence on robustness on data degradation of each fusion method. It is important to notice that all reported results were verified on separate test subsets, which remains unchanged among fusion methods.

### 4.1. Comparison of Fusion Method

In the first stage of experiments, we compared the performance of fusion methods on the BiGS dataset in the grasp outcome classification—a success or a failure. Input modalities were time series of gripper positions, orientations, and 3-axis forces from a wrist-mounted force-torque sensor. The final results were presented in Table 2. We reported the mean accuracy [%] with its standard deviation among the consecutive folds. The best performing models of Late (I-fold), MoE (III), Mid (II), and LMF (I) were chosen for the stage of experiments that includes the assessment of their robustness against data degradation and influence of input data augmentation. The fact that the average results in subsequent folds were very similar means that differences in data distributions across folds were negligible.

**Table 2.** The comparison of four fusion methods performed on the BioTac Grasp Stability Dataset (BiGS) dataset.

|      | I    | II   | III  | IV   | V    | Mean         |
|------|------|------|------|------|------|--------------|
| Late | 88.9 | 88.1 | 87.9 | 88.5 | 88.1 | 88.3 ± 0.4   |
| MoE  | 89.0 | 88.3 | 89.1 | 87.6 | 88.4 | 88.5 ± 0.6   |
| Mid  | 88.1 | 89.9 | 89.0 | 87.8 | 88.6 | 88.7 ± 0.8   |
| LMF  | 89.6 | 88.4 | 88.0 | 88.4 | 88.9 | 88.7 ± 0.6   |

Cross-validation on the HaTT dataset was another step in our experiments. Results in the form the classification accuracy [%] were reported in Table 3. As input modalities, we used again time series—a squashed 1-dimensional representation of acceleration and velocity, together with a normal force acting on a haptic device's tool-tip. For the next experiments, we chose the II-fold models for the Late, MoE, LMF methods, and III-fold model for the Mid fusion approach.

In the task of multi-label classification of haptic adjectives, we used the PHAC-2 dataset. Similarly, as in the [12], we chose as a performance metric the Area Under a Curve (AUC) that measures the area under the Receiver Operating Characteristic (ROC) curve. That metric is widely spread in the multi-label classification field of machine learning. It measures how good the predictive model can distinguish between classes (in our case—haptic adjectives) taking into account a correspondence between a sensitivity/specificity ratio and multiple values of a decision threshold. In the AUC-ROC metric, a value of 1.0 refers to an excellent classification ability, 0 means that the model is always wrong, while 0.5 means that model has no discrimination capacity. In Table 4 we reported AUC-ROC

metrics achieved by evaluated fusion methods. For further stages of experiments, we chose V-fold models of the Late and the MoE methods, I-fold of the Mid and IV for the LMF.

**Table 3.** The comparison of four fusion methods performed on the Penn Haptic Texture Toolkit (HaTT) dataset.

|      | I    | II   | III  | IV   | V    | Mean          |
|------|------|------|------|------|------|---------------|
| Late | 79.5 | 80.8 | 79.4 | 78.3 | 79.5 | 79.5 ± 0.9    |
| MoE  | 77.9 | 78.9 | 74.3 | 76.6 | 73.4 | 76.2 ± 2.3    |
| Mid  | 78.9 | 75.4 | 79.8 | 78.3 | 76.6 | 77.8 ± 1.8    |
| LMF  | 78.1 | 80.9 | 78.9 | 78.3 | 79.5 | 79.1 ± 1.1    |

**Table 4.** The comparison of four fusion methods performed on the PHAC-2 dataset. All values represent the Area Under a Curve (AUC)-Receiver Operating Characteristic (ROC) performance metric.

|      | I     | II    | III   | IV    | V     | Mean            |
|------|-------|-------|-------|-------|-------|-----------------|
| Late | 0.923 | 0.924 | 0.922 | 0.923 | 0.925 | 0.923 ± 0.001   |
| MoE  | 0.923 | 0.919 | 0.919 | 0.923 | 0.927 | 0.922 ± 0.003   |
| Mid  | 0.929 | 0.922 | 0.922 | 0.927 | 0.925 | 0.925 ± 0.003   |
| LMF  | 0.896 | 0.898 | 0.902 | 0.908 | 0.900 | 0.901 ± 0.005   |

## 4.2. Data Degradation Robustness

In the following section we present the results gathered from experiments regarding the robustness of selected methods against a variety of input data degradation scenarios. The research carried out brought very important conclusions on the capabilities of each fusion method to translate knowledge from one modality to another and revealed that in most cases there exists a phenomenon, which we called a leading modality. Namely, for each dataset there was a modality, the leading one, which regardless of the fusion method used is crucial for obtaining good results. To make this dependency visible, we presented our results in the form of heat-maps (see Figures 4–6).

In heat-maps, there were presented changes in a performance caused by decreasing the quality of one or more input modalities. We tested fusion methods against scenarios described below (a–e) and each row in heat-maps corresponds to one of the scenarios:

(a)　N—a normal noise $N$ added to selected modalities with a 0 mean and 0.7 standard deviation;
(b)　U—a uniform noise $U$ added to selected modalities that varies in the range ($-0.5$ to 0.5);
(c)　0—setting zeros in place of selected modalities, what simulated a deactivated/broken sensor;
(d)　RN—replacing selected modalities with normal noise $N$;
(e)　RU—replacing selected modalities with normal noise $U$.

Each heat-map column was annotated by a number that specify affected modalities (e.g., by the added uniform noise). For each dataset we tested fusion methods using three input modalities numbered as follows:

(a)　BiGS—*1*: gripper positions, *2*: gripper spatial orientations, *3*: 3-axis force;
(b)　HaTT—*1*: normal force, *2*: squashed acceleration, *3*: squashed velocity;
(c)　PHAC-2—*1*: images, *2*: raw electrodes from the 1st sensor, *3*: raw electrodes from the 2nd sensor.

At first, we tested selected fusion methods on the BiGS dataset and visualized the results on heat-maps in Figure 4. The use of heat-maps enabled one to easily inspect the knowledge alignment and translation properties of each fusion method. To perform these tests, the best performing models from Table 2 (marked in blue) were used.

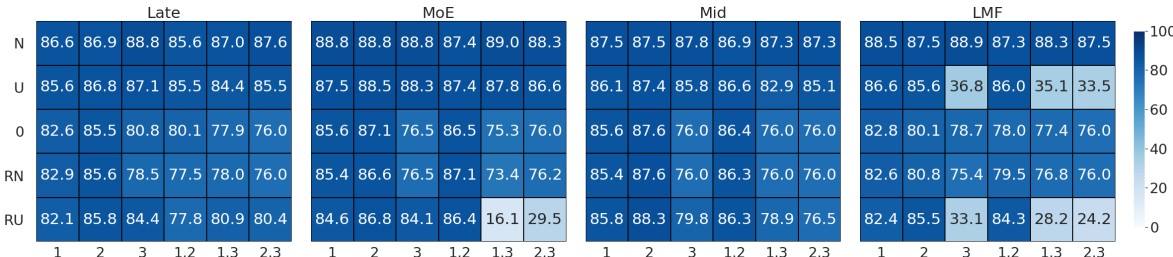

**Figure 4.** Results achieved by chosen models from the first stage of experiments on degraded data from the BiGS dataset. In heat-maps, there was presented a classification accuracy expressed in [%]. Rows correspond to different degradation scenarios, while columns are annotated by the indexes of affected modalities.

In Figure 5 we presented heat-maps generated for tests on the HaTT dataset. The influence of each modality on the final prediction is clearly visible and not every method is able to manage data degradation. Moreover, the 2nd modality (acceleration) appeared to be the leading one what resulted in a significant drop when it was noisy or faded. On the other hand, removing other modalities from the input data stream did not affect the final accuracy.

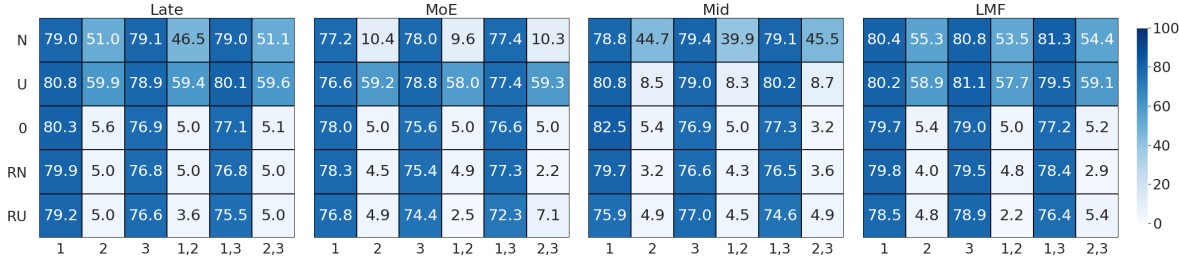

**Figure 5.** Accuracy [%] of a texture classification achieved while testing different fusion methods on degraded data from the HaTT dataset. Leading modality played a decisive dominant role, which resulted in a decreased quality in case of its degradation.

The AUC-ROC metric for the multi-label classification of haptic adjectives was reported in Figure 6. Similarly as in the experiments on the HaTT dataset, the leading modality is also visible. However, its correlations with other modalities played an even more important role in the final performance of methods. Inspecting heat-maps one can observe that the most meaningful correlations for predictions are between images (1st) and raw electrodes signals (2nd and 3rd). On the other hand. Noised interactions between both electrodes' time series only slightly influenced the classification performance.

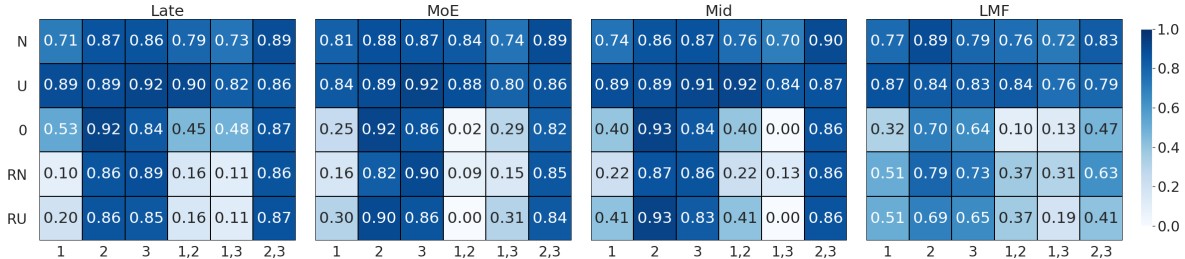

**Figure 6.** The AUC-ROC reported for the multi-label classification task using the PHAC-2 dataset. The leading modality is visible, but the correlations between modalities also affect predictions.

### 4.3. Data Augmentation vs. Leading Modality

Generated heat-maps from the previous stage of experiments revealed that in every dataset there exist one leading modality that had the biggest impact on the prediction. To verify whether the

degradation of the leading modality was a determining factor for the decreased performance of fusion methods we conducted more experiments using the data augmentation technique and removing a leading modality from the input data stream. In the following set of experiments we noised/faded the 33% of randomly selected training samples of a leading modalities in the training dataset. In the degraded part of the dataset, a half of samples were zeroed, while others were noised with the normal noise with the 0 mean and standard deviation set to 0.7. This value should be related the fact that standard deviation of the data was confined to one through data standardization described in Section 3.1. Again, to perform the experiments we chose models from the Section 4.1 marked with a blue color and re-trained them on the same folds as they were trained originally. As in previous experiments, all of the methods were tested again separate test subsets. Results were reported in Tables 5–7.

First of all, we re-trained fusion models on the augmented training dataset and tested them on the same versions of test datasets as in the Section 4.1. We did that to verify the impact of the data degradation on the performance on test sets without it. An accuracy [%] and AUC-ROC values [-] were presented in Table 5.

**Table 5.** Outcomes of multi-modal fusion methods obtained for models trained on datasets containing noised/zeroed inputs from the leading modality, but tested on the dataset without such samples.

|            | Late  | MoE   | Mid   | LMF   |
| ---------- | ----- | ----- | ----- | ----- |
| BiGS [%]   | 88.26 | 88.89 | 88.64 | 89.39 |
| HaTT [%]   | 78.15 | 75.8  | 75.1  | 76.95 |
| PHAC-2 [-] | 0.92  | 0.93  | 0.93  | 0.91  |

Secondly, we evaluated the performance of each fusion method on test datasets with the same proportion of the noised samples of the leading modality without zeroed samples. Results obtained during that trial were reported in Table 6.

**Table 6.** Results obtained for models trained on datasets containing noised/zeroed inputs from the leading modality and a noised leading modality channel during tests.

|            | Late  | MoE   | Mid   | LMF   |
| ---------- | ----- | ----- | ----- | ----- |
| BiGS [%]   | 88.14 | 88.64 | 88.51 | 89.26 |
| HaTT [%]   | 65.85 | 69.95 | 65.3  | 68.3  |
| PHAC-2 [-] | 0.88  | 0.86  | 0.88  | 0.84  |

Finally, the influence of zeroed leading modalities on fusion methods was assessed during the last stage of experiments. Table 7 contains accuracy and AUC-ROC metric results gathered on the test dataset, where the leading modality was switched-off.

**Table 7.** Results obtained for models trained on datasets containing noised/zeroed inputs from the leading modality and a zeroed leading modality during tests.

|            | Late  | MoE   | Mid   | LMF   |
| ---------- | ----- | ----- | ----- | ----- |
| BiGS [%]   | 85.77 | 88.76 | 86.27 | 89.01 |
| HaTT [%]   | 54.85 | 57.05 | 53.95 | 53.25 |
| PHAC-2 [-] | 0.23  | 0.24  | 0.21  | 0.61  |

## 5. Discussion

The discussion of our experiments was divided into two parts—the discussion on the performance of each method and the analysis of their properties. In our work, results revealed that methods most widely spread in robotics can successfully deal with multi-modal data, however, they exhibit a

fragility to missing/noised modalities, which resulted in decreased performance in such situations. That phenomenon might be considered as the main drawback of data-driven approaches to multi-modal fusion. However, as neural networks are included among data-driven methods, we were able to reduce to some extent the adverse impact of input quality degradation using the data augmentation technique.

*5.1. Comparison of Fusion Methods*

We compared the performance of four multi-modal fusion methods on three datasets including different time series (accelerations, velocities, forces), spatial transformations, and images. The important thing to note is that our target was to examine methods on the possibly largest set of homo and heterogeneous signals, hence we used different sets of modalities from each dataset and no modality was repeated between datasets even though, e.g., in the BiGS and PHAC-2, there were used the same tactile sensors—BioTacs.

The mean accuracy of all methods tested on the BiGS dataset was 88% and the differences between them were insignificant. Nevertheless, the most efficient fusion method in the grasp classification task was the LMF due to a smaller standard deviation among folds than the second method—the Mid fusion. The HaTT tests exhibited a slight increase of a mean results variance among different methods and standard deviations among folds comparing to BiGS results. In the texture recognition classification based on haptic signals, the best method turned out to be Late fusion, achieving a mean test accuracy of 79.5% between folds and additionally the smallest standard deviation equal to 0.9%. In the multi-label classification of haptic adjectives based on visual and haptic data the Late, MoE, and the Mid fusion methods were extremely close to each other in terms of a mean AUC-ROC metric, achieving a result of 0.92. The LMF turned out to be marginally below the performance represented by other methods.

On all tested datasets, the results from 5-fold cross-validation and tests on separate subsets showed that data used in experiments was consistent and there were no significant outliers among folds. That finding made it possible to carry out reliable experiments and ensure a fair comparison. Mean values of performance metrics among different fusion methods may suggest that the type of data fusion affects the performance of the model only to a very small extent. When all modalities are available and free of any noise it seems that more important was a reliable data preparation (e.g., ensuring a balanced distribution between classes in the train and test subsets, as well as between folds) for the training procedure than the fusion algorithm itself. In that section, apart from the comparison of different methods, we trained and chose the best-performing models for the next experiments. Chosen models were marked with blue color in Tables 2–4. All of the tested neural networks were trained in an end-to-end manner and performed relatively well, exhibiting a great capacity to learn from large (BiGS, HaTT) and small (PHAC-2) datasets. Further tests were conducted towards an assessment of the impact of each modality on the final result, and verifying the robustness of each method against input data degradation.

*5.2. Data Degradation Robustness*

All experiments from Sections 4.1–4.3 revealed the existence of a leading modality, which means that in our experiments there was always a one modality that played a dominant role for the discrimination between classes.

In Figure 4, one can observe that, for BiGS dataset, MoE and LMF fusion methods exhibited significantly decreased performance when the modality no. 3, which was a 3-axis force signal in combination with other modalities, was replaced by the uniform noise. Hence, we consider this signal as a leading modality in the BiGS dataset. As it can be observed, the LMF and MoE were also sensitive to the uniform noise added that affected the force signal. However, the described phenomenon did not occur for other fusion methods—the Late and Mid. They appeared to be relatively robust for data degradation scenarios, exhibiting no more than 10% of a drop in the accuracy when the leading

modality was noised, zeroed or totally replaced by noise. In a case of MoE the drop in the quality of discrimination between classes to the 16.1% and 29.5% would be caused by the fact that during the training, the gating network was trained to increase the importance of a leading modality for the prediction because it played a dominant role in provided data. Thus, when that signal, together with any of the other modality, were replaced by the noise, too much emphasis was put on the correlation of these two signals, what resulted in prediction mistakes. However, MoE exhibited robustness for any other data degradation scenarios. It appeared to be sensitive mainly to the correlations between the force signal and two other modalities—hand positions and orientations. Henceforth, in the MoE, the interactions between the modalities were essential for the grasp outcome classification, not the leading modality itself. Similar results were achieved while testing the LMF method—when the force signal was replaced by the uniform noise, the method was more likely to make mistakes, thus the interactions between modalities were meaningless. Additionally, the LMF was sensitive to not only scenarios with replacing the leading modality with noise and its combinations with other modalities, but also for the noise added to the signals. This could be observed in the LMF heat-map in Figure 4 looking at the results (36.8%, 35.1%, and 33.5%) in the *U*-row. The described phenomenon might be caused due to the fact that LMF is a tensor-based method that highly relies on outer products between uni-modal representations inside networks, thus the highest emphasis was put on inter-modality interactions. When a finding of these interactions was difficult/not possible, the LMF struggled to find a correct prediction for the grasp outcome evaluation task. It can be also observed that the type of noise introduced to the input data played a significant role for the prediction performance because presented findings were not observed for a normal noise. To explain that effect, we speculate that during the training phase, in signals there was already some noise present similar to the normal level, which resulted in a higher robustness for such a data degradation. The achieved robustness was truly substantial and it appears that a balance between the importance of modalities was paramount. Nevertheless, verifying that relevance is very challenging and involves a great number of experiments.

Another dataset involved in experiments was the HaTT, and results gathered during that trial were reported in Figure 5. Contrary to the BiGS dataset, by inspecting heat-maps one can observe that the 2nd modality (an acceleration) caused a significant drop of the accuracy for all tested data degradation scenarios and fusion methods. Hence, we consider an acceleration to be a leading modality from proposed set of input modalities. In the task of texture classification based on haptic signals, all methods exhibited a similar performance and sensitivity on different disturbances. A lack of a legitimate acceleration signal (zeroing or replacing with a noise) always caused a decreased performance to the level of 5% for all tested methods. This means that in the proposed set of modalities, the domination of an acceleration was tremendous, and the rest of the signals did not provide meaningful information about the process under investigation. The late fusion and LMF heat-maps evaluation gave a similar results for all data degradation scenarios, but the MoE and Mid fusion differs in terms of managing the added noise—MoE exhibited sensitivity to an appearance of the noise component in the leading modality, but the Mid was fragile for the same phenomenon but with the uniform noise added.

The results of the multi-label classification of haptic adjectives performed on the PHAC-2 dataset were shown in Figure 6. The biggest drop in the performance metric was reported for columns missing the 1st modality—an image, which was considered as a leading modality. In the MoE, the fact that the lack of images was able to cause a total failure of the classifier achieving the result of 0 (which means that the modal was always wrong) again indicates that gating network during a training put too much emphasis on the dominant modality. Taking into account that every result below 0.5 level means that the classifier is more often wrong than right on average. The Late, MoE and Mid fusion methods behaved similarly across all scenarios—the performance without a leading modality was significantly decreased. The LMF performed slightly different, achieving relatively good results when an input image was replaced by a noise what can be seen in the first column of the LMF heat-map. However, it performed worst in case of scenarios where other modalities were replaced by a noise/zeroed.

Although, it should be noted, that it does not achieved 0 AUC-ROC as it happened in case of MoE and Mid methods. Additionally, sometimes one can observe an improvement in the classification performance achieved when one modality was noised/faded. Such a phenomenon was reported, e.g., for the Mid fusion when the 2nd modality (raw electrode signal) was zeroed or replaced by a uniform noise. Comparing to Table 4 the improvement was 3%.

*5.3. Data Augmentation vs. Leading Modality*

In the last stage of experiments we re-trained models from the Section 4.1. We did that on the same folds as in the original experiment to ensure a fair comparison and provide results comparable with performed comparison of fusion methods.

First of all, we augmented the training dataset by adding normal noise $\mathcal{N}(0, 0.7)$ or switching-off the leading modality in 33% of the samples. Then, we re-trained selected models on corresponding folds and reported a mean accuracy and AUC-ROC metric of the classification for test datasets in Table 5. By measuring the influence of the data augmentation we established a point of reference for further experiments, as such augmentation could decrease the performance. As we can observe, we obtained similar results as in Tables 2–4, which means that this partial degradation does not affect the performance in nominal conditions significantly. In Tables 6 and 7 we reported the results for the degraded test set. In Table 5 the results from tests for noised leading modality were presented, whereas in Table 7 those for zeroed modality were presented.

As it can be observed in both tables, the data augmentation procedure increased robustness on noised and missing modalities entirely for the BiGS dataset, thus all methods gave similar results as during the tests on the data without any degradation applied on an leading modality. We believe that a proposed data augmentation procedure is sufficient to ensure a robustness on noised/missing samples for the proposed set of input modalities.

However, the above statement is not always true, which is clearly visible in results obtained for the HaTT dataset, when the mean decrease of accuracy was from 6% to 13% when comparing Table 5 to Table 6 and even larger from 18% to 24% between Tables 5 and 7. The results proved the same conclusions as before—the leading modality in the HaTT dataset possessed so much information meaningful for the discrimination between textures and other modalities played only a supporting role for that task. Nevertheless, using data augmentation still brought a significant improvement in results comparing to data degradation scenarios showed in Table 5. In both tested variants, the best performing method turned out to be the MoE, which indicates that the gating network learned to more efficiently refuse a predictions based on a degraded leading modality.

In the multi-label classification task on the PHAC-2 the Late, Mid and MoE methods failed to properly assign haptic adjectives when the vision was missing. However, the LMF method apparently was able to find intra- and inter-modality interactions that led to the surprisingly good result of 0.61 AUC-ROC metric. It indicates that the LMF was the only method that was able to actually assign the haptic adjective properly more often than make a mistake on average. In tests involving noise-only samples, all methods achieved similar result and the performance metric dropped only by 4–6%.

## 6. Conclusions

In our work we compared four multi-modal fusion methods that could be regarded as state-of-the-art. We assessed their performance in the three tasks—a prediction of a grasp outcome, a texture recognition and multi-label classification of haptic adjectives. Then, selected methods were verified in the variety of possible scenarios of input data degradation that might occur in real life, e.g., a sensor turn-off or a measurement noise. Finally, we measured the influence of data augmentation technique on the predictive capabilities of tested methods and again evaluated their robustness on noise added to the leading modality and its zeroing. We hope that the findings contained in our paper will make researchers realize that State-of-the-Art fusion methods are prone to over-fit to specific modalities, so-called leading modalities, and are rather susceptible to the noise as well as sensor

failures. Thus, in order to build reliable autonomous systems, we have to focus more on the robustness of our data fusion methods. Due to that, all our code and data used in experiments were made available as open-source.

**Author Contributions:** M.B. and P.K. conceived and designed the experiments; M.B. performed the experiments; M.B. and P.K. analyzed the data; M.B., P.K. and K.W. wrote the paper. All authors have read and agreed to the published version of the manuscript.

**Funding:** This work was supported by grant No. LIDER/3/0183/L-7/15/NCBR/2016 funded by The National Centre for Research and Development (Poland). M.B and K.W. were partially supported by the PUT Faculty of Control, Robotics and Electrical Engineering grant SBAD/0207 in year 2019.

**Conflicts of Interest:** The authors declare no conflict of interest.

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
