# Peer review of "On Robustness of Multi-Modal Fusion—Robotics Perspective"

_electronics, doi:10.3390/electronics9071152_

Round 1
Reviewer 1 Report
This paper focused on robustness of multi-modal fusion, which was rarely considered in others. Four multi-modal fusion methods were compared and several ex[eriments were conducted. This work is more than meaningful and intertesting. But some questions as follows are needed to be answered.
1). The motivations of this work should be explained clearly, thus the Introduction should be improved. Maybe the section Related work should be integrated into the section Introduction.
2).Four fusion methods were chosed in this work, and the authors should present the chosen rules clearly.
3).English language and style are minor spell check required
Reviewer 2 Report
Question 1: This method can be used for controlling the robotics to alleviate disturbances. There are many methods to solve this issue such as Disturbance observer, Data fusion from Kalman Filter,…However, the author did not mention in this paper. What are the main advantages of this research in comparison with other mentioned methods, the author should be more clearly explain in the introduction part?
Question 2: As mentioned in Question 1, there are some researches which should be mentioned in the article to make a rich of overview picture such as “An Extended Multi-Surface Sliding Control for Matched/Mismatched Uncertain Nonlinear Systems Through a Lumped Disturbance Estimator”, IEEE Access, Vol.8, pp. 91468-91475. “Robust Dynamic Sliding Mode Control-Based PID–Super Twisting Algorithm and Disturbance Observer for Second-Order Nonlinear Systems: Application to UAVs” Electronics 2019, 8(7). “Finite-Time Control of Multirotor UAVs Under Disturbances”. IEEE Access, pp.173549-173558.
